# Transcriptomic Characterization of Cow, Donkey and Goat Milk Extracellular Vesicles Reveals Their Anti-inflammatory and Immunomodulatory Potential

**DOI:** 10.3390/ijms222312759

**Published:** 2021-11-25

**Authors:** Samanta Mecocci, Daniele Pietrucci, Marco Milanesi, Luisa Pascucci, Silvia Filippi, Vittorio Rosato, Giovanni Chillemi, Stefano Capomaccio, Katia Cappelli

**Affiliations:** 1Department of Veterinary Medicine, University of Perugia, 06123 Perugia, Italy; samanta.mecocci@studenti.unipg.it (S.M.); luisa.pascucci@unipg.it (L.P.); stefano.capomaccio@unipg.it (S.C.); katia.cappelli@unipg.it (K.C.); 2Sports Horse Research Center (CRCS), University of Perugia, 06123 Perugia, Italy; 3Department for Innovation in Biological, Agro-Food and Forest Systems (DIBAF), University of Tuscia, 01100 Viterbo, Italy; daniele.pietrucci.89@gmail.com (D.P.); marco.milanesi@unitus.it (M.M.); 4Institute of Biomembranes, Bioenergetics and Molecular Biotechnologies, IBIOM, CNR, 70126 Bari, Italy; 5Department of Ecological and Biological Sciences, University of Tuscia, 01100 Viterbo, Italy; silvia.filippi@unitus.it; 6Genechron Srl, Via Giunio Antonio Resti 63, 00143 Roma, Italy; vittorio.rosato@genechron.com

**Keywords:** milk, extracellular vesicles, mEVs, RNAseq, miRNA, anti-inflammatory, immunomodulatory, cow, donkey, goat

## Abstract

Milk extracellular vesicles (mEVs) seem to be one of the main maternal messages delivery systems. Extracellular vesicles (EVs) are micro/nano-sized membrane-bound structures enclosing signaling molecules and thus acting as signal mediators between distant cells and/or tissues, exerting biological effects such as immune modulation and pro-regenerative activity. Milk is also a unique, scalable, and reliable source of EVs. Our aim was to characterize the RNA content of cow, donkey, and goat mEVs through transcriptomic analysis of mRNA and small RNA libraries. Over 10,000 transcripts and 2000 small RNAs were expressed in mEVs of each species. Among the most represented transcripts, 110 mRNAs were common between the species with cow acting as the most divergent. The most represented small RNA class was miRNA in all the species, with 10 shared miRNAs having high impact on the immune regulatory function. Functional analysis for the most abundant mRNAs shows epigenetic functions such as histone modification, telomere maintenance, and chromatin remodeling for cow; lipid catabolism, oxidative stress, and vitamin metabolism for donkey; and terms related to chemokine receptor interaction, leukocytes migration, and transcriptional regulation in response to stress for goat. For miRNA targets, shared terms emerged as the main functions for all the species: immunity modulation, protein synthesis, cellular cycle regulation, transmembrane exchanges, and ion channels. Moreover, donkey and goat showed additional terms related to epigenetic modification and DNA maintenance. Our results showed a potential mEVs immune regulatory purpose through their RNA cargo, although in vivo validation studies are necessary.

## 1. Introduction

Other than being a valuable nutrition source, milk represents a sophisticated signaling system that delivers maternal messages to the offspring [1]. This property seems to be mostly mediated by molecules enclosed in micro/nano-sized membrane-bound structures called extracellular vesicles (EVs) [2]. EVs are involved in cell-cell communication by transporting DNAs, mRNAs, microRNAs, lipids, and proteins to close and distant cells or tissues, protecting their labile content against degradation and providing a vehicle for cargo uptake by recipient cells. They can exert a plethora of biological effects with immunomodulating, anti-inflammatory, anti-cancer and pro-regenerative activities [3,4,5,6,7,8,9,10,11,12,13,14,15]. Signals can be as simple as an antigenic stimuli by EVs proteins or more complex as epigenetic modifications mediated by their miRNA cargo [16].

Food can be considered a valuable source for the mass production of EVs for their multiple uses; since milk is a widely available and inexpensive raw material for their production [17], milk-derived EVs (mEVs) have emerged as a reliable and scalable EVs source [18,19,20,21,22,23]. EVs from colostrum and breast milk were first isolated in 2007 in the human species [23], then subsequent studies reported the isolation from other mammals (cow, camel, buffalo, pig, and sheep) [24,25,26,27,28]. Milk immunomodulatory and anti-inflammatory activity, mainly due to the miRNA content [29,30], has been proven for human, cow, and donkey [31,32].

It is interesting that EVs and their RNA content are unaffected by acidification of the gastric environment [24,33] and different types of cells can uptake EVs from human and bovine milk [20,29,34]. For example, porcine milk EVs and their miRNAs are internalized by IPEC-J2 intestinal epithelial cells, modifying target gene expression and promoting intestinal cell proliferation [25]. Recent evidences demonstrate that milk EVs and miRNAs, taken up by human intestinal cells, can also reach the systemic circulation modifying gene expression of distant cells [35,36]. Moreover, besides the determination of high conservation level in miRNA coding genes through different species, these studies suggest a broad gene regulation in recipient cells, since each one of them targets several protein-coding genes [18,25,37,38,39].

MiRNAs adsorption through EV endocytosis after milk ingestion may play a role in the regulation of innate and adaptive immunity even through epigenetic changes [30,40,41]. Milk miRNA-mediated epigenetic regulation is a physiological phenomenon representing a natural channel for genetic material transfer to offspring and milk consumers [40]. Milk miRNAs also seem to be positively involved in resistance to allergy development and to autoimmune diseases [29,30].

Several studies have shown that miRNAs are associated with inflammatory processes in chronic diseases, regulating several mechanisms involved in inflammation initiation and resolution [42,43]. Within this path, the role of EV-mediated immune responses in the pathogenesis of chronic inflammation, such as inflammatory bowel disease (IBD), was recently investigated [44]. Non-coding RNAs play a pivotal role [45,46], taking part in the regulation of inflammatory processes, immune system modulation, and in the mucosal and microbiota homeostasis [44]. IBD, including Crohn’s disease (CD) and ulcerative colitis (UC), is a chronic relapsing inflammatory disorder of the intestine of unclear etiology that has become a global disease with an increasing incidence in newly industrialized countries [47].

In addition, there are evidences that miRNAs themselves can be regulated by feeding, suggesting that diet manipulation could be a promising therapeutic approach for modulating the risk of chronic diseases [31,48,49,50].

Moreover, the comparative study of mEVs in domestic species, while highlighting the particular cellular messages in different milks, might help to understand how the selection of mutational changes in miRNAs and/or miRNA binding sites could have provided a mechanism to generate some of the traits that differentiate domesticated species [51].

With this work, we aimed to complete our journey [52] in characterizing the molecular content of cow, donkey, and goat mEVs through a comprehensive RNA analysis to search for future applications as a nutraceutical in inflammatory conditions such as IBD.

## 2. Results

A complete overview of the different experiments and analyses carried out in the study is shown in Figure 1, wherein the upper part of the figure represents the milk sampling, the EV isolation method, and the morphological analysis with the relative synthetic representation of results. Below, the two bioinformatic pipelines used for RNA sequencing data analysis are detailed in the two arrows: the orange one for mRNAs and the green one for small RNAs. The references to figures and tables generated from the different analytical steps and representing the results obtained from this work are reported between the two arrows.

### 2.1. Morphologic Characterization of mEVs

The presence and purity of mEVs in the pellet generated from the milk of the three species along with size range and shape assessment were determined, respectively, by Western blotting, Transmission Electron Microscopy (TEM), and Nanoparticle Tracking Analysis (NTA). Figure 2 shows the electron micrographs, the size distributions observed via NTA, and the antibody reaction against mEV antigens.

At TEM, mEVs appeared rather homogeneous in shape and showed mainly intact limiting membrane. They ranged from 30 to 150 nm and were sometimes arranged in aggregates. No cell debris and minimal background were detected, revealing the efficiency of the purification procedure (Figure 2a). Based on the NTA data, the mEV mean (±standard error) diameter was 142.7 ± 2.9 nm for cow, 150.5 ± 3.2 for donkey, and 124.1 ± 2.3 for goat (Figure 2b). The distribution of mEV populations was “gaussian-like” for all the species with a unique peak and a narrow range (cow: D10 = 92.6 ± 1.7 nm and D90 = 215.2 ± 4.1 nm; donkey: D10 = 98.3 ± 1.7 nm and D90 = 219.7 ± 4.7 nm; goat: D10 = 75.9 ± 1.6 nm and D90 = 184.5 ± 3.5 nm), demonstrating homogeneity both within and between species. NTA also determines nanoparticle densities, reporting the mean concentration (particles/mL) (±standard deviation) of five measurements after a pellet resuspension in 400 μL of PBS that resulted as quite similar for the three specie: 1.22 × 10^12^ (±3.63 × 10^10^) for cow; 3.51 × 10^11^ (± 1.22 × 10^10^) for donkey; and 7.39 × 10^11^ (±1.57 × 10^10^) for goat. Western blot assay proved the mEV presence, showing a positive reaction for Tumor Susceptibility gene 101 protein (Tsg101) in all samples. Cluster of Differentiation 81 (CD81) expression was positive in donkey and goat, but not in cow (Figure 2c).

### 2.2. Molecular Characterization of mEVs RNAs

#### 2.2.1. Messenger RNAs

A total of 326 million raw reads were generated from single-end sequencing of three replicates per species on Illumina platform with an average of more than 36 million reads per sample. Over 99% of the produced sequences were retained after quality control and trimming procedures; of these, 84.8–90.9% were uniquely mapped on the reference sequences. Mapping statistics details are reported in Appendix A.

To further confirm mEV isolation purity and exclude cell contamination, a gross comparison between the number of expressed genes (reads per kilobase per million mapped reads-RPKM > 0) in mEV and mammary glands, from publicly available studies, was conducted. Only bovine mRNAs were used since udder mRNA sequencing data for the other two species are not yet available. The mean number of expressed genes in mEVs (14,407.67 ± 339.37) was significantly lower (*p*-value = 0.0286) than mammary glands (19,117.25 ± 1424.25). At the same time, the mean expression profile was significantly different (*p*-value < 2.2 × 10^−16^) between mEVs (1.16 ± 4.09) and mammary glands (0.48 ± 4.56). A graphical representation of the two gene sets is available in Appendix A.

Gene body coverage analysis carried out through the RSeQC package revealed a good representation in almost the entire transcript set length both in cow and donkey while goat showed a general reduction at the 3′-end (Appendix A).

A total of 10,656, 9464 and 11,143 expressed genes (RPKM > 1) were found for cow, donkey, and goat, respectively. For the differential expression analysis between mEV mRNAs of the three species through DESeq2 software, a core of 5907 orthologous genes (with a one-to-one type of orthology) was taken into account. Pairwise comparisons were applied, showing cow as divergent species compared to the others (Figure 3) with twice as many up-regulated genes (DEGs are reported in Appendix A).

For each species, transcripts found to be up-regulated in both the pairwise comparisons were considered enriched (Appendix A) and selected for functional analysis of the three gene ontology (GO) vocabularies (Appendix A). All of these enriched transcripts showed an RPKM value equal to or greater than six in at least one species, assessing a relevant expression level. The 78 GO terms significantly enriched in cow are representative of 19 functional groups, including the regulation of telomere maintenance and histone modifications, ncRNAs transport, and regulation of cytokines stimulus response. Terms of lipid catabolism, oxidative stress, and vitamin metabolism were enriched for donkey among the 63 GO terms enclosed in 13 functional groups. Goat showed 28 GO terms representative of 11 functional groups including terms related to chemokine receptor interaction, leukocytes migration, histone modification, and transcriptional regulation in response to stress. All species show significant enrichment in GO terms related to microtubule function. All GO details are reported in Appendix A.

#### 2.2.2. Small RNAs

Concerning small RNA features in cow and goat, almost all the reads were aligned to the microRNA class (miRNAs), comprising more than over 99% of total RPKM. Although this was the most representative class also in donkey (57% of total RPKM), the remaining portion of sequences fell into miscellaneous RNA, mostly Y-RNA and Vault classes (Appendix A).

In Table 1 are reported the higher expressed miRNAs (chosen from the features covered by 95% of total RPKM, Appendix A) resulting in 41, 28, and 40 miRNA for cow, donkey, and goat, respectively. As shown in Appendix A, 10 of these are shared among the three species. Details on sharing and uniqueness are provided in Table 1.

MiRNAs listed in Table 1 were used as input for MiRWalk analysis, obtaining their validated targets differentiated for target binding sites (3′-UTR, 5′-UTR, and CDS) (Table 2). However, for chi-miR-3431, eca-miR-3548, bta-miR-3600, bta-miR-669, bta-miR-148c, and bta-miR-148d it was not possible to identify the human homolog. Therefore, they were excluded from the target analysis. For each species, a unique target list was generated and filtered according to the number of miRNA hits (five for cow and goat and four for donkey) (Appendix A).

Filtered targets were used to generate a Protein-Protein Interaction Network (PPI) using the IMEx database: a total of 123 nodes with 253 edges were produced for cow genes, donkey had 1025 nodes and 2272 edges, while goat showed 988 nodes and 2899 edges. Among the numerous protein clusters identified by the analysis, we selected those with an interaction threshold greater than 25, resulting in three clusters for cow and and eight for both donkey and goat (Figure 4). Briefly, a cluster is characterized by a central node, corresponding to the protein with the highest number of interactions with other proteins correlated to similar biological functions.

MDM4 was a central node protein for all the species, while donkey and goat shared an additional central node, SOCS4 (Figure 4; Appendix A). A GO enrichment analysis was then carried out on these clusters through the ClueGO app and results are reported in Appendix A for cow, donkey, and goat clusters, respectively.

## 3. Discussion

In the last two decades, extracellular vesicles (EVs) gained global attention, tickling researchers minds for new ideas and potential applications thanks to their intrinsic messenger carrier role [53]. In this study, the total RNA content of EVs isolated from cow, donkey, and goat milk (mEVs) was characterized, and their potential functional role highlighted, both for the mRNA and miRNA cargo. A focus was placed on their anti-inflammatory and immunomodulant potential as a conditional prerequisite to use mEVs for attenuating symptoms in inflammatory bowel diseases (IBD).

Three different species were chosen to evaluate the mEV RNA cargo since domestication and selection may have influenced metabolic phenotypes in modern livestock and during the domestication process, as pointed out in an in silico analysis [51].

One of the major flaws that one can encounter in the characterization of EVs content is based on the quality of the isolated material. Indeed, as a prerequisite for the RNAs analysis, we verified that our samples were essentially enriched in mEVs, ruling out protein and cellular contamination as reported by the TEM analysis (Figure 2a). All considered parameters such as shape and the dimensional range as well as the positive reaction to known markers (CD81 and Tsg101) corroborated a correct mEV isolation (Figure 2). Moreover, a check on sequenced reads was then carried out, comparing mEV mRNA RPKM with sequencing data of mammary glands deposited on SRA database. As expected, a reduction in the number of expressed genes was observed in mEV RNA (Appendix A).

Then, analyzing the mEV RNA cargo, the total number of genes with an RPKM greater than one was quite similar for the three species, with approximately 10,000 transcripts. This number is in line with what previously observed for cow [54], porcine [55], and rat [56] mEVs; however, differences in RNA integrity were observed.

Although precise functions of EV mRNAs in target cells have not been clarified, it is known that complete mRNAs or fragmented mRNAs that keep the 5′ starting codon can be transferred in the receiving cell and directly translated into protein, modifying the global protein balance and consequently influencing metabolic pathways and biological processes [57,58]. On the other hand, previous experiences show widespread mRNA fragmentation mainly affecting the 5′-end of the typical transcript from mEVs [54] and from EV mRNA in general [59,60], hypothesizing a regulatory role (competitor, regulator of stability, localization, and translational activity of mRNAs) in target cells [60,61].

In detail, we detected high-quality mRNA in our mEVs, especially for cow and donkey with a peak at the 5′-end, maintaining good coverage for the rest of the sequence and a slight reduction for the very 3′-end (Appendix A). Goat mEVs were characterized by a general higher degradation while maintaining a good representativity at the 5′-end.

Then we evaluated if genes enclosed in mEVs were differentially expressed among the three species, although a direct comparison of expression levels is not trivial. For this reason, we implemented a procedure based on orthologous genes and proper normalization. The resulting pairwise comparisons show similarity between donkey and goat (Figure 3), while cow have a higher up-regulated number of genes (DEGs, reported in Appendix A). The greatest part of the selected orthologous genes (5907) was stable in terms of expression, but species signatures emerged. Cow, for example, showed 875 and 1055 up-regulated genes compared to donkey and goat, respectively. The donkey had 561 over-expressed genes with respect to cow and 468 with respect to goat. Finally, 615 and 371 genes in goat were found over-expressed with respect to cow and donkey. These data may suggest that the three cargos deliver different messages to the target cells. The GO results, where different terms for each species emerged (Appendix A), seem to corroborate this hypothesis. In cow, one of the major functionalities for mEV mRNAs is related to epigenetic regulation and DNA preservation, with enrichment of terms related to histones acetylation, regulation of mRNA metabolic process, ncRNA export from the nucleus, tRNA modification, and regulation of telomere maintenance. Another enriched GO term in cow, “Positive regulation of stress-activated MAPK cascade”, points towards regulatory properties. Ideed, this is a crucial signaling pathway for cell homeostasis, which activates the transcription of various regulatory proteins [62]. For donkey, an activity focused on the vitamin metabolism and the fatty acid catabolism, with terms linked to peroxisome and oxidation, appeared to be enriched. In line, human and camel milk-derived EVs have been proven to improve oxidative stress conditions in pathological environments [63]. “CCR chemokine receptor binding” was the main functional group of GO terms (i.e., with the highest number of associated GO terms and genes) in goat, indicating immune system cells as a possible source of these mEVs. Milk EVs, indeed, are highly heterogeneous and originate from a variety of cell populations residing in the source organism’s mammary gland, such as immune cells [64].

Concering our data for small RNAs, only donkey showed an important fraction of Y-RNAs. Although this finding could be biased by a reduced assembly quality of the ass genome and a suboptimal annotation, many read counts were attributed to some Y-RNAs. These small ncRNAs have recently gained interest due to the discovery of their abundance in some EVs types, especially released in biofluids, showing some kind of involvement in the immune system [65].

Concerning miRNAs, the majority of molecules highly expressed and shared between the three species (let-7i, miR-148b, miR-151a, miR-186, miR-191, miR-200b, miR-21, miR-27b, miR-30b, and miR-30d) are known to be enriched in milk [20,66,67,68] and involved in immunomodulation [22,63,67,68,69,70,71,72,73]. Although several studies have investigated the miRNA content in milk EVs from individual species [67], very few are about EV-associated miRNAs among different species [68]. Interestingly, in all studies, several abundant miRNAs were shared between species and are coherent with our results. These miRNAs were implicated in immune-related functions, regulation of cell growth, and signal transduction [68].

The most abundant miRNA in breast, cow and goat milk is miR-148, which is highly conserved in mammals [66,68]. Moreover, miR-151 and miR-186 were proposed as potential breast milk biomarkers [73]. MiR-148 is consistently reported in virtually all milk small RNA sequencing, supporting its evolutionary importance in lactation and for the newborn healthy development [74,75].

Regarding the most expressed common miRNAs in mEVs from all species and focusing on the potential immunomodulatory and anti-inflammatory activity, Liu and coauthors found that miR-151 interacts with *STAT3* mRNA, inducing down-regulation in mouse macrophages in response to LPS stimulation, and amplifying the initial innate immune response [71]. In addition, it is also known that miR-151 suppresses the expression of Th1 cytokines such as IL-2, IL-12, and IFN-γ [76].

MiR-148 regulates DNA methyl-transferase 1 (DNMT1), suggesting that milk consumption might affect the epigenome signatures [40,67] through mEVs uptake. Interestingly, milk miR-148a-3p and miR-29a-3p can also downregulate DNMT3B [77], a protein responsible for Forkhead box P3 (FOXP3) epigenetic inhibition [78]. Therefore, by preventing its epigenetic inhibition by DNMT3B, milk miR-148a and miR-29a could enhance *FOXP3* expression, driving the differentiation of T lymphocytes towards the anti-inflammatory regulatory T cell phenotype. This effect could be an explanation for mEVs’ beneficial role in rheumatoid arthritis [79]. Recent research linked milk-derived miRNA-148a to pancreatic beta-cell differentiation, promoting a potential protective role against type 2 diabetes mellitus development [80]. The biological effects of these miRNAs corroborate the mRNA enrichment findings already described with processes involved in the epigenetic modification and DNA maintenance.

Regarding other common mEV miRNAs, let-7 together with miR-148, regulate transcription factor NF-κB in vivo, with the immune response suppression as an outcome [67,68]. Moreover, miR-30b and miR-200, along with miR-148, have been designated as major immune-related miRNAs [81] and are highly represented in milk [82] and also present in our core molecules in mEVs (Table 1).

MiR-200a-3p [83] also promotes the proliferation of intestinal epithelial cells through the epithelial-mesenchymal transition inhibition via TGF-β [84] that, as recently reported, exerts a pleiotropic effect towards the immune network balance via SMAD activities in the nucleus [85]. SMAD transcription factors act through the transforming growth factor-beta/SMAD (TGF-β/SMAD) signaling pathway that sets the balance between immune activation for pathogen clearing and the immune suppression to limit damage to self [85]. Since SMAD members are increased in Crohn’s disease patients [85] and are validated targets in our analyses (Appendix A), the mEV message (miRNA content and their targets) could acquire particular importance in the IBD pathogenesis.

MiR-21, another most abundant small RNA in our mEVs, beyond its presence in milk, is one of the most expressed members of the small non-coding microRNA family in many mammalian cell types; its expression is enhanced in many diseases and in inflamed tissues. MiR-21 is part of a complex regulatory feedback cascade where it can negatively regulate the pro-inflammatory response induced by different stimuli; in particular, miR-21 has emerged as a key mediator of the anti-inflammatory response in macrophages and a potential marker of immune cell activation [69].

In addition to the shared miRNAs, we found many peculiar miRNAs in each species (Table 1, Appendix A). Concerning cow, many miRNAs were found to have a pivotal role in inflammation, such as miR-340 which targets JAK, cyclin D1, and MMP2, that in turn down-regulate cytokine production [86]. This miRNA is often downregulated during the development of autoimmune diseases. Moreover, miR-340 directly targets the pro-inflammatory cytokines IL-4 [87] and IL-17A [88] in psoriasis mice model, where Th17 cells and the consequent IL-17A production are implicated in the pathogenesis of this autoimmune disease as in others such as IBD [89]. Similar effects seem to be addressed to miR-181a with anti-inflammatory effects mediated by the down-regulation of IL-1α [90] and the inhibition of NF-κB activation [91]. Furthermore, miR-99a can reduce macrophage M1 phenotype and increase M2 activation by targeting TNFα [92]; miR-182-5p can modulate the TLR4/NF-κB signaling pathway, inhibiting the inflammatory response induced by LPS treatment [93], while miR-429 modulates the IL-8 production through the NF-κB inhibition, reducing inflammation in epithelial cells [94]. Therefore, it was proposed as a candidate for anti-colitis therapy thanks to its capability of modulating mucin secretion in human colorectal cells and mouse colitis tissues [95].

An ability to modulate many pro-inflammatory cytokines such as IL-6, IL-1β, and TNFα in different tissues was shown by miR-26a, miR-27a, and miR-125b that we found most abundant in goat mEVs [86]. Concerning abundant miRNAs in donkey, miR-320a through its targets modulates the NF-κB signaling in IBD, while miR-23a and miR-145 inhibit TLR4 and IL-17 signaling, decreasing IL-6, MCP-1, and metalloproteinases production [86].

Target genes and protein network analyses (Figure 4, Appendix A) on highly expressed miRNAs in each species underline the EV message finalized to protein synthesis and cellular cycle regulation (Appendix A). Moreover, we retrieved donkey and goat enriched GO terms involved in epigenetic modification and DNA maintenance (donkey cluster 1 “chromatin maintenance”, “chromatin organization”—donkey cluster 5 “histone methyltransferase binding”, “regulation of gene expression, epigenetic “—goat cluster 4 “positive regulation of histone H3-K4 methylation “—goat cluster 2 “ATP-dependent chromatin remodeling”) (Appendix A). GO terms that refer to transmembrane exchanges and ion channels were found for all species (cow cluster 1 “calcium-dependent protein binding”, “positive regulation of intracellular transport”—goat cluster 5 “calcium-dependent protein binding”, “positive regulation of intracellular protein transport”—donkey cluster 6 “calcium-dependent protein binding”, “positive regulation of intracellular transport”) (Appendix A). These categories are congruent with those of mEV mRNA GO analyses.

In addition, peculiar mEV miRNA targets appeared to be linked to immunity (cow cluster 1 “CD40 receptor complex”—goat and donkey cluster 1 “I-kappaB kinase/NF-kappaB signaling”—donkey cluster 1 ”positive regulation of NF-kappaB transcription factor activity”, “positive regulation of leukocyte apoptotic process” ”interleukin-12-mediated signaling pathway”—donkey cluster 2 “T cell costimulation”, “regulation of B cell proliferation”—donkey cluster 5 “antimicrobial humoral immune response mediated by antimicrobial peptide”—goat cluster 1 “response to cytokine”, “positive regulation of tumor necrosis factor-mediated signaling pathway”—goat cluster 2 “interleukin-2 production”) and signal transduction (cow cluster 3 “adenylate cyclase-modulating G protein-coupled receptor signaling pathway”, “phospholipase C-activating G protein-coupled receptor signaling pathway”—donkey cluster 2 “AMP-activated protein kinase activity”—goat cluster 3 “cAMP response element binding”—goat cluster 9 “signaling receptor complex adaptor activity”) (Appendix A).

The abundance of the molecules both present and targeted reinforce the hypothesis that mEVs may attenuate intestinal inflammation in IBD conditions [93], although in vivo validation studies are necessary.

Indeed, while not directly assessed here as this is a characterization study, mEV RNA cargo has promising potential biological effects as recent literature comfirmed that transported miRNAs in EVs are partially stable during gastrointestinal digestion, and bioavailable since they are capable of reaching target tissues [96]. While this is intriguing, it will be important to improve research in the minimum amount of transcripts for impacting gene expression, the possible degradation within the gastrointestinal tract, the amount that might reach the circulation and the cell to exert a biological effect, which are all aspects that have been poorly investigated in in vitro/in vivo experiments [68,97].

## 4. Materials and Methods

### 4.1. Milk Collection

Three samples for each species (cow, goat, and donkey) were collected from mass milk, to cope with the individual variability. Milk was sampled from farms surveilled by the Veterinary Medicine Department, University of Perugia: the Didactic Zootechnical Farm of the University for bovine and two distinct owned farms for donkey and goat. The goat milk was sampled from animals derived from a crossbreed of Camosciata delle Alpi and Saanen; donkey milk was collected from crossbreed animals. In both the cases, samples were taken during the mid-lactation period and animals were fed with grass, cereals, and hay in extensive pastoralism fashion. Cow milk was collected from Holstein Friesian cattle breed farm, where the greatest part of the animals was in mid-lactation. Rearing conditions are referable to standard intensive farms, with unifeed and mechanical milking. Milk was immediately processed or stored at 4 °C for less than 24 h before, avoiding cryo-preservation to minimize artifacts.

### 4.2. EVs Isolation

Milk EVs were isolated by serial differential centrifugations (DC) and a step in ethylenediaminetetraacetic acid tetrasodium salt dihydrate (EDTA), following the protocol of Vaswani and collaborators [98], with little modifications hereafter reported. In brief, preliminary centrifugation steps were used to eliminate fat globules on the surface and cellular debris and protein complexes in the pellet, recuperating the intermediate phase. Three hundred (300) mL of each milk were subjected to two consecutive 3000× *g* centrifugations for 10 min at room temperature (Eppendorf^®^ Centrifuge 5810R with a F34-6-38 rotor); then, an equal volume of 0.25 M EDTA (pH 7.4) was added to the supernatant, incubated for 15 min on ice, and centrifuged at 10,000× *g* for 1 h at 4 °C. Then, centrifugation at 35,000× *g* for 1 h at 4 °C was carried out using polyallomer tubes in a Beckman Coulter Optima L-100 XP with an SW41 Ti rotor. A final ultracentrifugation at 200,000× *g* for 90 mins at 4 °C was carried out collecting mEVs in the 24 separate pellets used for morphological analysis and RNA characterization.

### 4.3. Milk EVs Characterization

#### 4.3.1. Western Blotting

RIPA Buffer (20–40 μL) was added to a pellet for each species to extract proteins, incubating the solution for 15 min in ice. The cell lysates were centrifuged at 13,000 rpm for 5 min and the supernatant containing the proteins was recovered, according to Botta et al. 2019 [99]. Proteins were quantified at Qubit^®^ 3.0 fluorometer (Thermo Fisher Scientific, Waltham, MA, USA) and 25–30 μg protein samples were analyzed by immunoblotting on 10% SDS-PAGE as previously described [100]. CD81 (bs-6934R, Bioss Antibodies, Woburn, MA, USA) diluted 1:500 and TSG-101 (sc-7964, Santa Cruz Biotechnology, Santa Cruz, CA, USA) diluted 1:400 were used as primary antibodies. A HRP-conjugated IgG was used as secondary antibody (Vector Labs, San Francisco, CA, USA) diluted 1:10000. The signal was detected using the enhanced chemiluminescence method following the manufacturer’s instructions (Amersham) using Chemi Doc XRS system (Bio-Rad Laboratories Ltd., Hemel Hempstead, UK) and images processed with ImageLab (BioRad Laboratories Ltd., Hemel Hempstead, UK).

#### 4.3.2. Transmission Electron Microscopy (TEM) and Nanoparticle Tracking Assay (NTA)

A drop of mEV suspension (one pellet) was placed on Parafilm. mEVs were allowed to adere to the surphase of a Formvar-coated copper grid (Electron Microscopy Sciences) placed on the top of each drop for about 20 min. Grids were then washed in PBS and distilled water and then contrasted with 2% uranyl acetate for 5 min. The observation was performed using a Philips EM208 transmission electron microscope equipped with a digital camera (University Centre of Electron Microscopy—CUME). 

A Malvern Panalytical NanoSight NS300 nanoparticle tracking analysis (NTA) system (Malvern, Worcestershire, UK) was used to assess the concentration and size distribution of isolated mEVs. One mEV pellet of the three species was resuspended and diluted in filtered (0.22 μm pore size) phosphate buffered saline (PBS) (Sigma, St. Louis, MI, USA) to be suitable for the NTA system’s working concentration range and five measurements were performed for each one. Concentration and diameter results are reported as mean ± 1 standard error of the mean.

### 4.4. RNA Extraction and Library Preparation

To obtain a sufficient RNA amount suitable for sequencing procedure, 21 mEV pellets, obtained as described in 4.2, were immediately treated with 100 μL (each) of TRIzol™ (Thermo Fisher Scientific, Waltham, MA, USA) and total RNA was extracted through the miRNeasy Mini Kit (QIAGEN, Germantown, MD, USA) following the manufacturer’s instructions. After RNA extraction, samples were subjected to a DNase digestion using the TURBO DNA-freeTM Kit (Thermo Fisher Scientific, Waltham, MA, USA), then quantified through the NanoDrop 2000 spectrophotometer (Thermo Fisher Scientific, Waltham, MA, USA) and quality tested by the Agilent 2100 Bioanalyzer RNA assay (Agilent technologies, Santa Clara, CA, USA). The RNA extracted from mEVs was subjected to Next Generation Sequencing for both mRNA and small RNA; for each species, three libraries were prepared for both. Universal Plus mRNA-Seq kit (library type: fr-secondstrand) (Tecan Genomics, Redwood City, CA, USA) and QIAseq miRNA library kit (QIAGEN, Germantown, MD, USA) were used for the two libraries’ preparation following the manufacturer’s instructions. Final libraries were checked with both Qubit 2.0 Fluorometer (Invitrogen, Carlsbad, CA, USA) and Agilent Bioanalyzer DNA assay and sequenced on single-end 75 bp mode on NextSeq 500 (Illumina, San Diego, CA, USA).

### 4.5. Bioinformatic Analysis

#### 4.5.1. From Sequencing to Datasets

Raw reads (deposited in the Sequence Read Archive (SRA), see reference submission number at the Data Availability Statement section of the article) were checked for quality through the FastQC tool (https://www.bioinformatics.babraham.ac.uk/projects/fastqc/ accessed on 1 October 2021) and adaptors trimmed using TrimGalore (https://github.com/FelixKrueger/TrimGalore accessed on 1 October 2021). STAR aligner [101] was used for mRNA reads mapping to the relative reference genomes (ARS-UCD1.2 for cow, ARS1 for goat) using Ensembl annotation (release 102) [102], while ASM303372v1 downloaded from the National Center for Biotechnology Information (NCBI) website was used for donkey. FeatureCounts [103] was used to generate the count matrix on which the differential gene expression analysis was carried out.

For small RNA reads, BowTie2 [104] was used, setting parameters for short sequences and performing a two-step alignment procedure: a first passage on miRBase (v.22) [105] database and a second to reference the genome to retrieve accurate information on miRNAs and other small RNAs with the unmapped sequences from the first passage. Concerning miRBase database, the hairpin version of the species annotated sequences were used for cow and goat, while, for donkey, the horse annotation was adopted. A single count matrix was produced, merging the information generated from the two alignments through the count sum of the same miRNAs if covered by reads for both the miRBase and the genomic mapping. Uniquely mapped reads were selected for normalization of mRNA and smallRNA count matrices through the reads per kilobase per million mapped reads (RPKM) method and only features with an RPKM greater than 1 were considered for downstream analysis. This threshold was chosen to be as inclusive as possible in this step, preferring further filtering in downstream analyses.

Cow mRNA mEV profile was compared to mRNA data of mammary glands downloaded from the SRA database [106] (BioSamples: SAMN12831050, SAMN12831049, SAMN14600526 and SAMN14600525). The number of genes expressed (i.e., RPKM > 0) in the two samples was compared (Wilcoxon rank sum one-tail test). A frequency histogram comparing the average expression profile, expressed in log_2_(RPKM) at intervals of 0.5 was generated. A statistic test (Wilcoxon rank sum test) was conducted. RNA quality in terms of transcript integrity was evaluated through RSeQC package [107].

#### 4.5.2. Differential Gene Expression Analysis between the Three Species

For cargo mRNAs of mEVs, a differential expression analysis was carried out, basing on a unique list of orthologous genes, selecting those with the one-to-one type of orthology available for all the three species. The RPKM matrix was log-2 transformed and it underwent a normalization process that allows the comparison of gene expression levels across different species. The normalization process consists of the calculation, for each sample, of median-scaling factors across the 1000 most conserved genes. Then, the scaling factors were used to normalize all the genes in all samples [108,109]. The scaling factors were computed by using the normalization function available at the GitHub repository [108]. After the normalization process, data were squared to handle normalization data with RPKM values < 0. Finally, statistical analyses to identify all differential express genes were performed in R 3.5.3 using the DESeq2 package [110]. Using the contrast parameter in the results function, we evaluated all the pairwise comparisons (cow vs. donkey, cow vs. goat, and donkey vs. goat). For each comparison, only genes with a False Discovery Rate (FDR) less than 0.05 (FDR < 0.05) and a log_2_ Fold Change (FC) less than −2 or greater than 2 (|log_2_ FC| > 2) were considered as differential expressed genes. The R package EnhancedVolcano was used to produce the volcano plots for all the comparisons [111]. For each species, a list composed of the up-regulated genes found to be over-expressed in both comparisons with the other two species was generated. Since the list of genes used for differential analysis was created with a relaxed threshold, the risk of taking into account scarcely expressed genes was real. To overcome this, the initial RPKM relative to “one-to-one” transcripts was verified before proceeding with the functional analysis: all the RPKM values were equal to or greater than 6 in at least one species, ensuring relevant and reliable expression level for the functional analysis. These transcripts were used for gene ontology (GO) enrichment analysis through the ClueGO [112] application of Cytoscape (v. 7.1) suite [113].

#### 4.5.3. miRNA Targets Retrieving and Functional Analysis

For the most expressed miRNAs (covered by 95% of total RPKM) of the three species, human orthologues were retrieved through miRBase (v.22) and used to identify all the validated target genes. In order to select unique molecules for each miRNA, we traced back the portion sequenced in our experiment (5p or 3p) and the human homolog was used as input in the MiRWalk 3.0 (http://mirwalk.umm.uni-heidelberg.de/ accessed on 1 October 2021) database. A unique list of all targets for each miRNA was produced, specifying also the miRNA site of action on the targeted mRNA: 3′-UTR, 5′-UTR, or CDS (coding region). The targets were filtered for the number of miRNA hits, taking into account those genes targeted by five miRNAs for cow and goat and four miRNAs for donkey. The different miRNA hits threshold between donkey and the other two species is explained by the lower number of most expressed miRNAs in this species compared to the others. A Protein-Protein Interaction Network (PPI) was build starting from targets through the Cytoscape 3.7.1 suite [113], using the IMEx database [114], which contains non-redundant information deriving from the major public protein databases. Then, the clusterMaker 2.0 app [115] with the “gLay” option was used to highlight different clusters within the network based on the number and type of connections between the nodes. Gene Ontology (GO) enrichment analysis was carried out on clusters with a number of interactions greater than 25 through the ClueGO application [112], and results were filtered for a FDR < 0.05 (Benjamini Hockberg correction).

## 5. Conclusions

This study allowed the in-depth characterization of mRNA and small RNA content of cow, goat, and donkey mEVs. Key molecules relevant for immune and inflammatory signals regulation and biological processes involved in transcription regulation and cell homeostasis were found, denoting similar functions for the mEVs of the three species. However, some peculiar features emerged. In particular, goat and donkey showed as the main functions those related to transmembrane ion channels and gene transcription, especially epigenetic regulation. A particular role in lipid metabolism and response to oxidative stress appeared for donkey mEVs.

The anti-inflammatory and immunomodulatory potential of cow, donkey, and goat mEVs highlighted in this study due to the characteristic RNA cargo is prodromal for further investigations through in vitro and in vivo models of inflammatory-based diseases such as IBD.

## Figures and Tables

**Figure 1 ijms-22-12759-f001:**
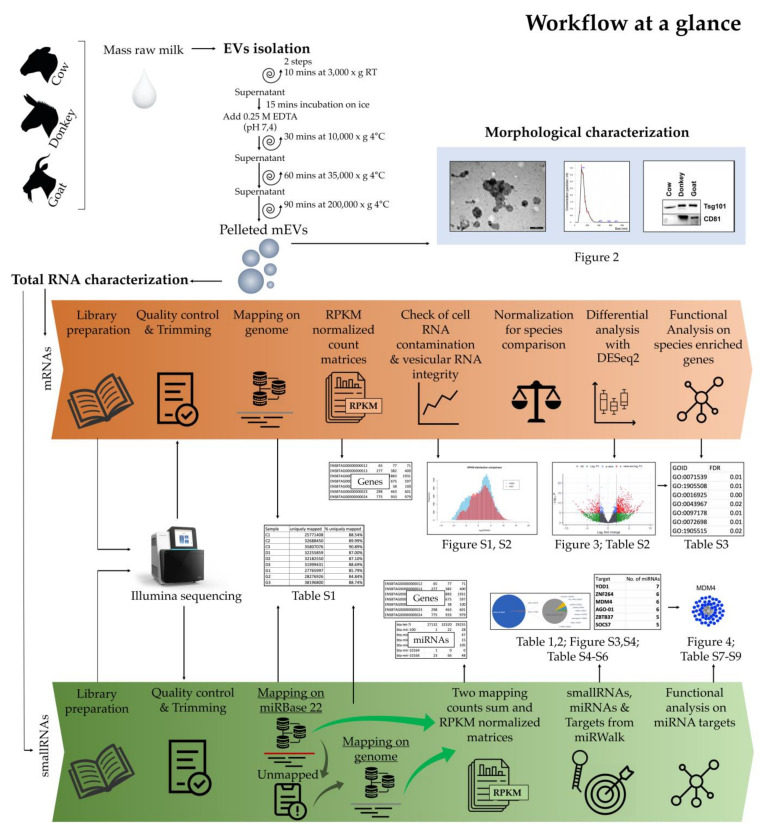
Workflow overview cartoon.

**Figure 2 ijms-22-12759-f002:**
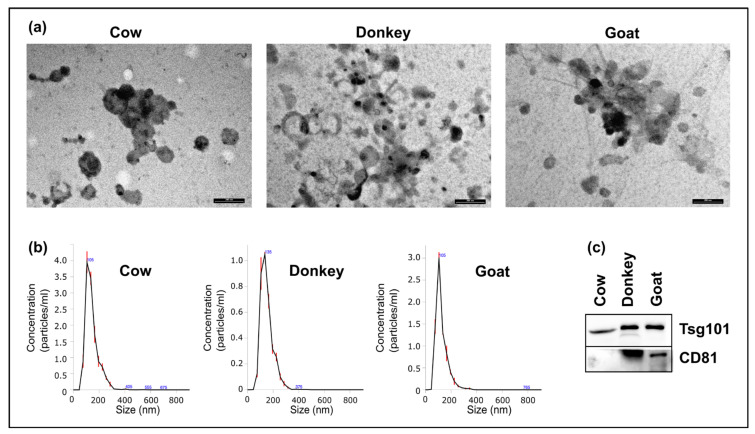
Morphological characterization of mEVs: (**a**) Transmission electron microscopy (TEM) revealed the presence of single and aggregated vesicles mainly in the range of 30–150 nm. Scale bar: 200 nm; (**b**) Size distribution of vesicles measured using the Nanosight NS300 nanoparticle tracking analysis system. The peak of the mEV size distribution was 105 nm for cow and goat and 135 for donkey; (**c**) Western blot results of Tsg101 (Tumor Susceptibility gene 101 protein) and CD81 (Cluster of Differentiation 81) tested mEV antigens.

**Figure 3 ijms-22-12759-f003:**
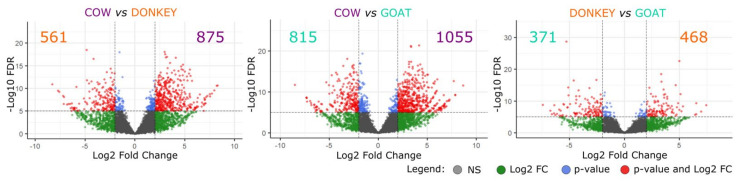
Volcano plots of differentially expressed genes through mEV RNA pairwise comparisons of the three species using DESeq2 for data analysis. The X-axis represents log_2_ fold change (log_2_ FC) of the pairwise comparison, while the Y-axis indicates significance of differential expression (False Discovery Rate - FDR). The gray dots indicate no significant (NS) changes in the gene expression (|log_2_ FC| < 2, FDR > 0.05), the green dots denote genes with |log_2_ FC| > 2 but not significant (FDR > 0.05), light blue dots indicate very small changes in gene expression (|log_2_ FC| < 2), while red dots represent significantly up- and down-regulated genes (|log_2_ FC| > 2, FDR < 0.05). Purple numbers indicate the significantly up-regulated genes in cow compared to the other reported species, the orange numbers indicate those up-regulated in donkey, and the teal numbers refer to goat over-expressed genes.

**Figure 4 ijms-22-12759-f004:**
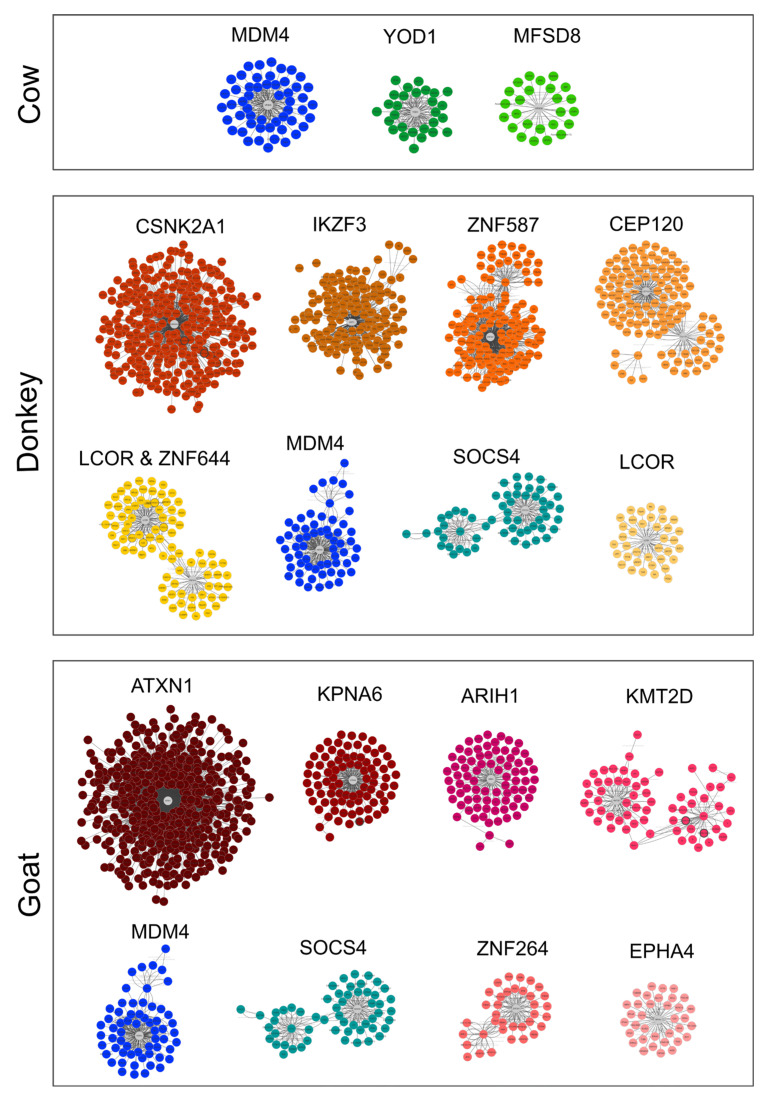
Clusters of proteins generated by clusterMaker 2.0 Cytoscape application starting from the PPI network. Central nodes (grey) indicate proteins with the higher number of interconnections. A consistent colour scheme was applied for clusters with the same central node in the different species.

**Table 1 ijms-22-12759-t001:** Most expressed miRNAs (covered by 95% RPKM) in mEV species cargo.

Enclosing mEVs	No. of miRNAs	miRNAs		
Cow–Donkey–Goat	10	miR-151a miR-191 miR-30b let-7i	miR-27b miR-30d miR-186	miR-200b miR-148b miR-21
Cow–Goat	15	miR-141 miR-23b miR-93 miR-25 miR-361	let-7b miR-660 let-7g miR-30a miR-200a	miR-125a miR-20a miR-29a miR-140 miR-200c
Donkey–Goat	6	miR-148a miR-143	miR-26b miR-30e	miR-223 miR-155
Cow–Donkey	1	miR-375		
Cow	10	miR-340 mir181a miR-362 miR-99a	miR-532 miR-26b miR-182	miR-429 miR-425 miR-34a
Donkey	10	miR-374a miR-152 miR-101 miR-98	miR-590 miR-320a let-7d	miR-23a miR-145 let-7c
Goat	8	miR-26a let-7f miR-27a	miR-22 miR-224 miR-16	miR-125b miR-190a

**Table 2 ijms-22-12759-t002:** Number of total and selected targets retrieved for each species.

Species	3′-UTR Targets	5′-UTR Targets	CDS Targets	Total Targets	Filtered Targets
Cow	1351	221	1196	2296	3
Donkey	814	109	603	1313	15
Goat	1537	256	1283	2568	14

## Data Availability

The raw sequencing data generated during this study are openly available in the Sequence Read Archive (SRA) at the National Center for Biotechnology Information (NCBI) servers under the Bio-Project PRJNA771627, with the following accession numbers: SAMN22322873, SAMN22322874, SAMN22322875, SAMN22322876, SAMN22322877, SAMN22322878, SAMN22322879, SAMN22322880, SAMN22322881, SAMN22322882, SAMN22322883, SAMN22322884, SAMN22322885, SAMN22322886, SAMN22322887, SAMN22322888, SAMN22322889, SAMN22322890.

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
