# Peer review of "Transcriptomic Characterization of Cow, Donkey and Goat Milk Extracellular Vesicles Reveals Their Anti-Inflammatory and Immunomodulatory Potential"

_ijms, 2021, doi:10.3390/ijms222312759_

Round 1

Reviewer 1 Report

Authors describes the transcriptomic characterization of milk EVs in cow donkey and goat. While the manuscript is technically sound and experiments are correct, they lack novelty and are merely descriptive. While previous studies have characterized small RNAs (miRNAs) content of these milk EVs, mRNAs characterization has been less studied.  A good point of this study is the comparative analysis among the three milk EV sources.

A section of limitations of the study should be incorporated. For example, in this section authors should indicate that all the analysis have been performed by bioinformatics and they do not necessarily represent the pathways that will be modified if someone (or a cell system) receive those milk EVs. Indeed, there is a large difference on what you find by bioinformatics and the possible biological effect that can be reached. There are several facts that will first need to be considered including: the amount of transcripts (from a quantitative point of view) that EVs can ferries, the degradation within the gastrointestinal tract (if consumed orally), the amount that might reach the circulation, the amount that might reach the cell and more importantly once taken up by the cell the amount of transcript that might reach within the cell and their localization in the specific part of the cell to exert a biological effect. All these obstacles and challenges should be clearly indicated in the limitations of the study. In this and other journal there are recent manuscript that study all these aspects and could be cited Int J Mol Sci. 2021 Jan 22;22(3):1105. doi: 10.3390/ijms22031105. Eur J Nutr. 2021 Oct 29. doi: 10.1007/s00394-021-02720-y.

Bioinformatic analysis was performed in mRNA expressed genes with an RPKM>1. This analysis is not stringent enough with and RPKM>1. This mean that authors used all expressed transcripts. To reach relevance in possible biological effects (assessed by bioinformatics, as described here) authors should re-analyse the data using mainly the most enriched transcripts. I would suggest increasing the cut off level of expression to RPKM largely higher than 1. This would reduce the number of transcripts per sample, but bioinformatic analysis will be more relevant for future studies. This new analysis might change the whole picture of results, but will be a good contribution of the research group.

Please explain, or modify the form of expressing small RNAs, as RPKM is more suitable for long transcript but not much for small RNAs (i.e. transcript of less than 50 bp). Why noy using only RPM?

Bioinformatic analysis of miRNAs should be only performed using validated targets genes. Predicted targets would give an unnecessarily high number of targets. Although miRNA targets can be also found at the 5’UTR and CDS canonical function of miRNAs target only the 3’UTR.

Abstract should include the potential limitations of the bioinformatic analysis. Authors could include something like: “..through their RNA cargo, although in vivo validation studies are necessarily.”

Author Response

  • A section of limitations of the study should be incorporated. For example, in this section authors should indicate that all the analysis have been performed by bioinformatics and they do not necessarily represent the pathways that will be modified if someone (or a cell system) receive those milk EVs. Indeed, there is a large difference on what you find by bioinformatics and the possible biological effect that can be reached. There are several facts that will first need to be considered including: the amount of transcripts (from a quantitative point of view) that EVs can ferries, the degradation within the gastrointestinal tract (if consumed orally), the amount that might reach the circulation, the amount that might reach the cell and more importantly once taken up by the cell the amount of transcript that might reach within the cell and their localization in the specific part of the cell to exert a biological effect. All these obstacles and challenges should be clearly indicated in the limitations of the study. In this and other journal there are recent manuscript that study all these aspects and could be cited Int J Mol Sci. 2021 Jan 22;22(3):1105. doi: 10.3390/ijms22031105. Eur J Nutr. 2021 Oct 29. doi: 10.1007/s00394-021-02720-y.

Response: A section of limitations was added in the text.

  • Bioinformatic analysis was performed in mRNA expressed genes with an RPKM>1. This analysis is not stringent enough with and RPKM>1. This mean that authors used all expressed transcripts. To reach relevance in possible biological effects (assessed by bioinformatics, as described here) authors should re-analyse the data using mainly the most enriched transcripts. I would suggest increasing the cut off level of expression to RPKM largely higher than 1. This would reduce the number of transcripts per sample, but bioinformatic analysis will be more relevant for future studies. This new analysis might change the whole picture of results, but will be a good contribution of the research group.

Response: We thank the reviewer to highlight this critical point that was probably poorly explained in the text. We choose to set RPKM>1 as a threshold for expressed mRNAs for two fundamental reasons: this work is based on the mEV RNA content deep characterization therefore being too stringent could have implied an important data loss; but the fundamental reason is that the downstream differential gene expression analysis between the 3 species was performed on orthologous genes exclusively (one-to-one type of orthology). The use of one-to-one orthologues is in itself an important limit.

Therefore, it was essential to start from as large a quantity of genes as possible. In this way, the presence of genes with a reduced level of expression allowed to highlight particularly expressed genes in only one species compared to the other two. Starting from a higher RPKM value could bring the risk of losing many genes expressed at high levels in one species and probably scarcely expressed in the others.

Indeed, the functional analysis was performed only on the "up-regulated" genes in one species with respect to the other two, likely with good levels of expression and which therefore may have the biological relevance mentioned by the reviewer. However, following the reviewer suggestion, we decided to carry out a downstream verification on the up-regulated DEGs for species (those used for functional analysis) and the corresponding RPKM values, finding an RPKM> 6 in the species in which the DEG was found up-regulated, a value that corresponds approximately to a hundred or more reads, a number that is far from irrelevant.

We thought that the reviewer's note was born from the lack of clarity regarding this choice, so we decided to point this out in the materials and methods by inserting the carried out verification.

  • Please explain, or modify the form of expressing small RNAs, as RPKM is more suitable for long transcript but not much for small RNAs (i.e. transcript of less than 50 bp). Why noy using only RPM?

Response: Although we agree with the reviewer's note regarding the use of RPKM for small RNAs, we thought to use this method in accordance with that used for mRNAs. We would like to maintain this method of normalization also for small RNAs as it does not constitute an error, from the scientific point of view. Normalizing for RPKM takes into account the length of the genes, although small RNAs have a constant length, applying this method brings neither benefit nor harm. Since RPKM has rightly been used for mRNA, we can think of using the same for small RNAs as well.

  • Bioinformatic analysis of miRNAs should be only performed using validated targets genes. Predicted targets would give an unnecessarily high number of targets. Although miRNA targets can be also found at the 5’UTR and CDS canonical function of miRNAs target only the 3’UTR. 

Response: Following the reviewer suggestion, miRNA targets analysis was repeated using validated targets exclusively. The materials and methods section as well as results and discussion were updated according to the new analysis.

  • Abstract should include the potential limitations of the bioinformatic analysis. Authors could include something like: “..through their RNA cargo, although in vivo validation studies are necessarily.”

Response: Done.

Reviewer 2 Report

The manuscript “Transcriptomic characterization of cow, donkey and goat milk extracellular vesicles reveals their anti-inflammatory and immunomodulatory potential” by Mecocci et al. is a research article in which the authors aim to characterize the RNA content in EVs isolated from cow, donkey and goat milk.

The work builds upon previous observations in which the same group of authors performed a metabolomic analysis of EVs from bovine, donkey and goat milk (Mecocci et al., 2020).

The presented manuscript is focused on an important and continuously evolving topic setting the grounds for the possible use of EVs for the treatment of several pathologies in virtue for their anti-inflammatory and immunomodulatory properties.

General Comment:

The main concern with the manuscript is that a mechanistic understanding of the RNA content from different sources is lacking and the study does not go very far towards establishing the specific biological functions of EV cargo towards the mentioned possible clinical applications.

Specific comments

- Line 7: “University of Perugia”; Line 11 “Università della Tuscia”. Keep consistent.

- Change font in figure 1

- Briefly illustrate Fig 1 in the text

- Line 104: “Figure 2 shows the antibody reaction again mEV antigens, the electron micrographs and the size distributions observed via NTA”. Describe in the text figure panels in the order they appear in the figure.

- Comment on different concentration observed in the different samples. How many replicates have been performed?

- Include a Venn diagram indicating the miRNAs from EVs from the 3 different sources

- Line 414: “The secondary antibody was).” Revise.

- Line 436: “Twenty-one (21) mEV pellets”. Please, specify better how you obtained and quantified these 21 pellets.

- You can refer to and compare in the discussion section data available on EV miRNA in human milk (Tingö et al. 2021).

- Similarly, results from studies which have previously compared abundance of miRNA in milk from different sources (such as van Herwijnen et al. 2018) should be discussed.

- Several environmental and genetic factors affect milk quality. How did you take these factors into account?

Reference:

Mecocci S, Gevi F, Pietrucci D, Cavinato L, Luly FR, Pascucci L, Petrini S, Ascenzioni F, Zolla L, Chillemi G, Cappelli K. Anti-Inflammatory Potential of Cow, Donkey and Goat Milk Extracellular Vesicles as Revealed by Metabolomic Profile. Nutrients. 2020;12(10):2908. doi: 10.3390/nu12102908.

van Herwijnen MJC, Driedonks TAP, Snoek BL, Kroon AMT, Kleinjan M, Jorritsma R, Pieterse CMJ, Hoen ENMN, Wauben MHM. Abundantly Present miRNAs in Milk-Derived Extracellular Vesicles Are Conserved Between Mammals. Front Nutr. 2018;5:81. doi: 10.3389/fnut.2018.00081.

Tingö L, Ahlberg E, Johansson L, Pedersen SA, Chawla K, Sætrom P, Cione E, Simpson MR. Non-Coding RNAs in Human Breast Milk: A Systematic Review. Front Immunol. 2021;12:725323. doi: 10.3389/fimmu.2021.725323

Author Response

- Line 7: “University of Perugia”; Line 11 “Università della Tuscia”. Keep consistent.

Response: Done.

- Change font in figure 1

Response: Done.

- Briefly illustrate Fig 1 in the text

Response: Done.

- Line 104: “Figure 2 shows the antibody reaction again mEV antigens, the electron micrographs and the size distributions observed via NTA”. Describe in the text figure panels in the order they appear in the figure.

Response: Done.

- Comment on different concentration observed in the different samples. How many replicates have been performed?

Response: A brief comment regarding the mEV concentration in the pellet of the three species together with the number of performed replicates was added in the results.   

- Include a Venn diagram indicating the miRNAs from EVs from the 3 different sources

Response: Done.

- Line 414: “The secondary antibody was).” Revise.

Response: Done.

- Line 436: “Twenty-one (21) mEV pellets”. Please, specify better how you obtained and quantified these 21 pellets.

Response: Added.

- You can refer to and compare in the discussion section data available on EV miRNA in human milk (Tingö et al. 2021).

- Similarly, results from studies which have previously compared abundance of miRNA in milk from different sources (such as van Herwijnen et al. 2018) should be discussed.

Response: We have added both these studies ad references and compare their results in the discussion section.

- Several environmental and genetic factors affect milk quality. How did you take these factors into account?

Response: We perfectly know that milk quality could be affected by the environment and genetics. At the same time, we want to remind you that the aim of this work is a preliminary in-depth characterization of RNA mEV content. For these reasons, we decided to start from mass milk samples to cope with individual variability, choosing the most popular and used breeds for milk production. Moreover, animals, that came from local farms monitored by the veterinary department of the University of Perugia, were reared and fed under canonical conditions. The next steps, starting from this work, could be a more specific characterization comparing the mEV RNA cargo between different breeds or different reared models or varying the diet, to evaluate as these features could influence the vesicular cargo.

Round 2

Reviewer 1 Report

The manuscript has been improved and now is suitable for publication.

Author Response

We thank the reviewer for suggestions that allowed the improvement of the manuscript.